# Aromatic-bridged and *meso-meso*-linked BF$_2$-smaragdyrin dimers exhibit fast decays in polar solvents by symmetry-breaking charge transfer

Lingbo Wen[1], Xian Cao[1], Seokwon Lee[2], Ling Xu[1], Yutao Rao[1], Seongsoo Kang[2], Dongho Kim [2✉], Atsuhiro Osuka[1✉] & Jianxin Song [1✉]

Symmetry-breaking charge transfer is one of the key process in photosynthetic reaction centers and specific artificial optoelectronic devices such as dye-sensitized solar cells. Here we report the synthesis of aromatic-bridged BF$_2$-smaragdyrin dimers, *meso*-free BF$_2$-smaragdyrin, and its *meso-meso*-linked BF$_2$-smaragdyrin dimer. The decays of S$_1$-states of these dimers are accelerated with an increase in solvent polarity and a decrease in the distance between the two BF$_2$-smaragdyrin units, suggesting symmetry-breaking charge transfer. The fluorescence lifetimes of the dimers become shortened in polar solvents. However, ultrafast transient absorption spectroscopy do not detect charge-separated ion pairs. On the basis of these results, we conclude that the decays of the excited states of the BF$_2$-smaragdyrin dimers are accelerated by solvation-induced symmetry-breaking charge transfer, depending on the degree of the electronic interaction between the smaragdryin units as a rare case for porphyrinoids. The degree of charge transfer is larger for dimers with larger electronic interactions.

[1] Key Laboratory of Chemical Biology and Traditional Chinese Medicine, Ministry of Educational of China, Key Laboratory of the Assembly and Application of Organic Functional Molecules of Hunan Province College of Chemistry and Chemical Engineering, Hunan Normal University, Changsha 410081, China. [2] Spetroscopy Laboratory for Functional π-Electron Systems and Department of Chemistry, Yonsei University, Seoul 03722, Korea. ✉email: dongho@yonsei.ac.kr; atsuhiroosuka@hunnu.edu.cn; jxsong@hunnu.edu.cn

In recent years, expanded porphyrins have emerged as attractive π-functional molecules in light of versatile electronic properties, rich coordination chemistry, and charming structures[1–5]. Among these, smaragdyrin had been possessing a legendary position, since its presence was reported along with sapphyrin by Woodward but had been long elusive until our first synthesis of 5,10,19-triarylsmaragdyrins in 2018[6]. Instead, Chandrashekar et al have extensively explored the chemistry of core-modified smaragdyrins to demonstrate that these porphyrinoids are promising in many fields including catalysts, molecular hosts, nonlinear optical materials, and coordinating ligands for large metal ions[7–11]. Our synthesis of 5,10,19-triarylsmaragdyrins **1** was based on a double nucleophilic aromatic substitution (SNAr) reaction of α,α'-dibrominated boron-dipyrromethene's (BODIPY's) with a 5,10-diaryltripyrrane[6]. Later, this synthetic protocol was used for the synthesis of singly and doubly Neo-confused smaragdyrins **2** and **3** (Fig. 1)[12].

In this paper, we report the synthesis of aromatic-bridged BF$_2$–smaragdyrin complex dimers **6a–6d** by similar SNAr reactions of aromatic-bridged α,α'-dibrominated BODIPY dimers **4a–4d** with 5,10-diaryltripyrrane **5**[13,14]. Further, we synthesized meso-unsubstituted BF$_2$-smaragdyrin **11** and succeeded in its oxidative dimerization with air to give meso–meso-linked BF$_2$–smaragdyrin dimer **12**. While several meso–meso-linked dimers of core-modified smaragdyrins and analogs were reported[15,16], this is the first example of meso–meso-linked smaragdyrin dimer.

Symmetry breaking charge separations (SB-CS) have been recognized to be important in relation to the charge transfer character of the bacteriochlorophyll special pair in the photosynthetic reaction centers, which is considered to trigger the sequential electron-transfer cascade[17,18]. But photoinduced SB-CS of artificial dimers composed of the same chromophores is rare[17–25], mainly because the energy gaps of the charge separation between the same chromophores are seldom negative enough to allow SB-CS. SB-CS has been never observed for porphyrins, while so many porphyrin dimers and oligomers were synthesized so far. Recently, an interesting example was reported for SB-CS in excitonically coupled subphthalocyanine dimer[26]. In this respect, smaragdyrin is an intriguing porphyrinoid that is active both in reduction and oxidation due to its expanded conjugated π-network. In this paper, we conclude that these smaragdyrin dimers revealed that the excited-state decays are clearly accelerated with an increase of the solvent polarity and a decrease of the distance between the two smaragdyrin units, suggesting the key

role of symmetry-breaking charge transfer in their decays as a rare case for porphyrinoids.

## Results

**Synthesis**. As the first targets, we synthesized aromatic-spacer-bridged BF$_2$–smaragdyrin dimers **6a–6d**. Building blocks **4a–4d** were obtained in moderate yields via a set of transformations including α-bromination of the corresponding spacer-bridged dipyrromethene dimers, and oxidation and complexation with BF$_2$ unit. Since the solubilities of **4a–4d** were very poor in common solvents and only acceptable in o-dichlorobenzene (o-DCB), the coupling reactions of **4a–4d** with tripyrromethane **5** were conducted in refluxing o-DCB in the presence of an excess amount of cesium carbonate for 48 h. Dimers **6a–6d** were isolated in low but reproducible yields of 2.8, 2.2, 2.7, and 2.4% yield, respectively (Fig. 2). These dimers were characterized by $^1$H and $^{13}$C NMR spectra and high-resolution matrix-assisted laser desorption ionization time-of-flight (MALDI-TOF) mass spectra (Supplementary Figs. 9, 11, 12, 14, 15, 17, 18, 20, 46–49). The structures of **6b** and **6c** were confirmed by X-ray diffraction analysis (Fig. 3). The crystal structure of **6b** shows that a dihedral angle between the BF$_2$–smaragdyrin units and the central 1,4-phenylene bridge is 47.3(2)°. This suggests that the 1,4-phenylene unit may serve as a moderate π-conjugation bridge for the BF$_2$-smaragdyrins units. The crystal structure of **6c** shows a dihedral angle of 51.7(2)° for the BF$_2$–smaragdyrin units and 4,4-biphenylene unit and a dihedral angle of 31.8(3)° for the two phenylene units in the 4,4'-biphenylene bridge. These structural features suggest that the 4,4'-biphenyl bridge serves as a weaker π-conjugation mediator.

The $^1$H NMR spectra of **6a–6d** in CDCl$_3$ are simple, reflecting the symmetric structures. Characteristically, signals due to the inner pyrrolic protons are downfield shifted as compared with the reference monomer **11**. On the other hand, signals of the β-protons (labeled as H$_1$, H$_2$, H$_4$, and H$_5$) are observed at nearly the same chemical shifts. Signals due to the bridging groups are down-field shifted, reflecting the diatropic ring current of the BF$_2$–smaragdyrin ring.

As the second target, we synthesized meso–meso-linked BF$_2$-smaragdyrin dimer **12**. Initially, preparation of α,α'-dibrominated and dichlorinated meso-free BODIPY's **8-Br** and **8-Cl** were attempted by halogenations of di(2-pyrroyl)methane **7** with NBS and NCS. But these attempts failed due to the instabilities of **8-Br** and **8-Cl**. We chose regioselective chlorination of BODIPY's with

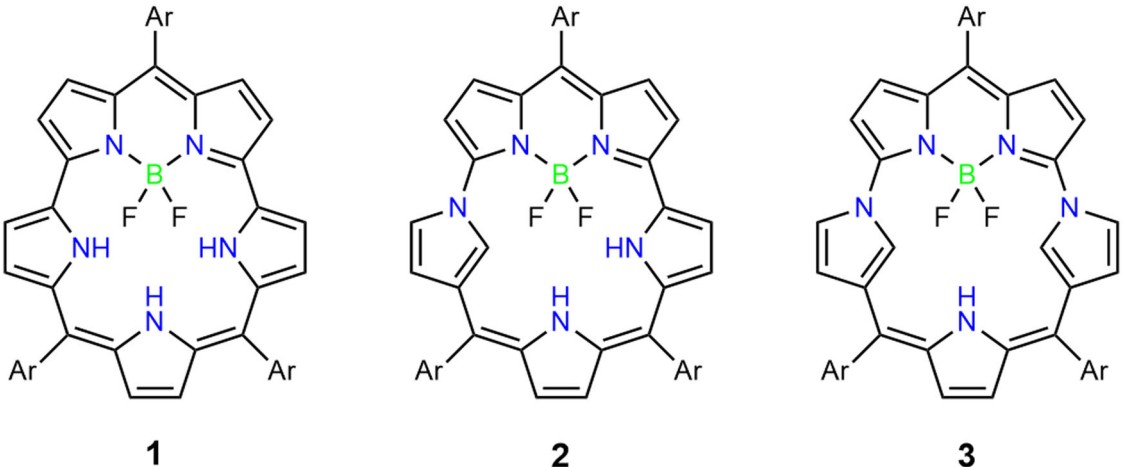

**Fig. 1 BF$_2$–smaragdyrin and Neo-confused BF$_2$–smaragdyrins.** Structures of BF$_2$–smaragdyrin **1**, singly neo-confused BF$_2$–smaragdyrin **2**, and doubly neo-confused BF$_2$–smaragdyrin **3**.

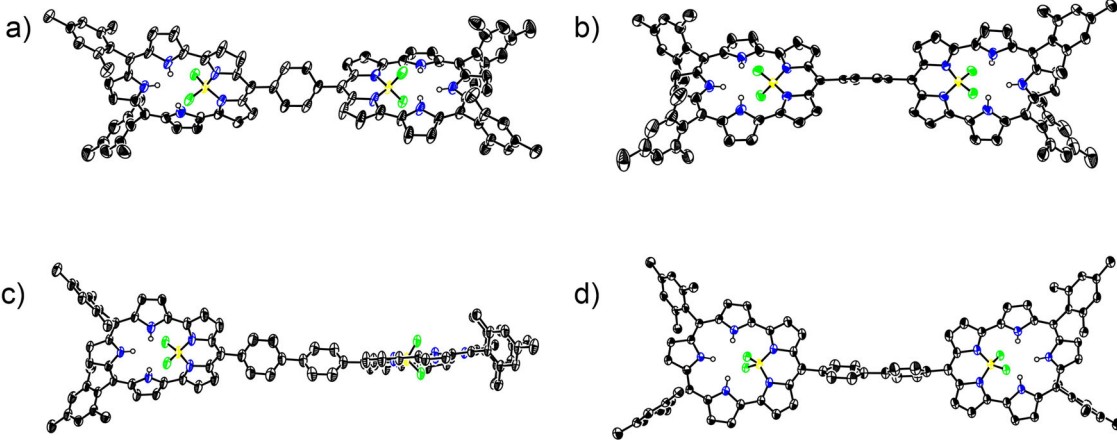

**Fig. 2 Synthesis of BF₂–smaragdyrins dimers.** Synthesis of **6a–d** through S$_N$Ar reactions.

**Fig. 3 X-ray structures of 6b and 6c. a** Top view and **b** side view of **6b**. **c** Top view and **d** side view of **6c**. Solvent molecules and hydrogen atoms on carbon have been omitted for clarity.

CuCl₂ developed by Jiao and Hall[27]. BODIPY **9** was prepared by oxidation of **7** with *p*-chloranil in CH₂Cl₂ at −40 °C under nitrogen followed by reaction with BF₃•OEt₂ in 38% yield. Then, the reaction of **9** with CuCl₂ and Cu(OTf)₂ under refluxing acetonitrile gave **10** in 55% yield (Fig. 4). The structure of **10** was fully confirmed by ¹H NMR, ¹³C NMR, HR-MS, and single crystal X-ray diffraction analysis (Fig. 4 and Supplementary Figs. 2, 3, 38, 51).

With **10** in hand, we tried to synthesize *meso*-diaryl-substituted BF₂–smaragdyrins by our method[6]. Namely, the reaction of **10** with 5,10-dimesityl-tripyrromethene **5** in the

presence of cesium carbonate in refluxing *p*-xylene overnight afforded 5,10-dimesityl-[22]smaragdyrin BF₂-complex **11** in 5.2% yield after usual work up (Fig. 5). The ¹H NMR spectrum of **11** in CDCl₃ displays a singlet due to the *meso*-proton at $\delta = 10.71$ and five signals due to the outer pyrrolic β-protons between $\delta = 10.46$ and 8.94 ppm, a singlet due to the inner NH proton at −7.76 ppm, and a triplet due to the inner NH proton at −4.90 ppm that is coupled with the two fluorine atoms through-space coupling with $J = 11.2$ Hz. The ¹H NMR spectrum of **11** is similar to that of the triaryl-substituted BF₂–smaragdyrin reported before[6], indicating its aromatic

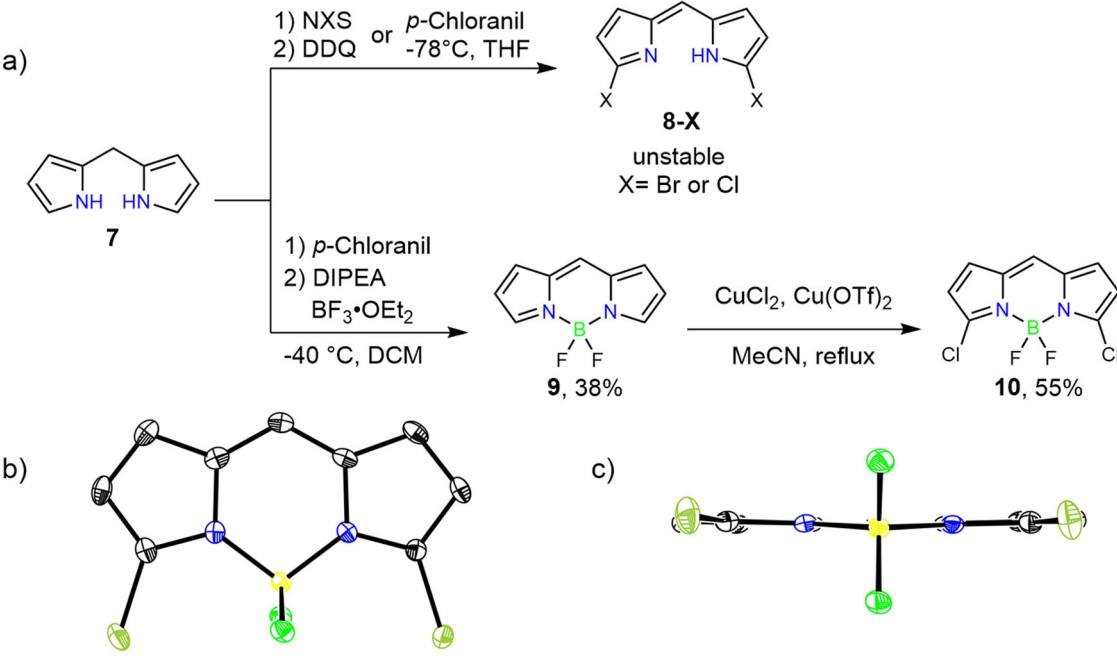

**Fig. 4 Synthesis and X-ray crystal structures of α, α′-Halogenated meso-free BODIPY 10. a** Synthesis of α, α′-dichloro meso-free BODIPY **10** and α, α′-dibromo/diiodo meso-free dipyrrin **8-X**. **b** Top view of crystal structure of **10**, and **c** side view of crystal structure of **10**. Solvent molecules and hydrogen atoms on carbon have been omitted for clarity.

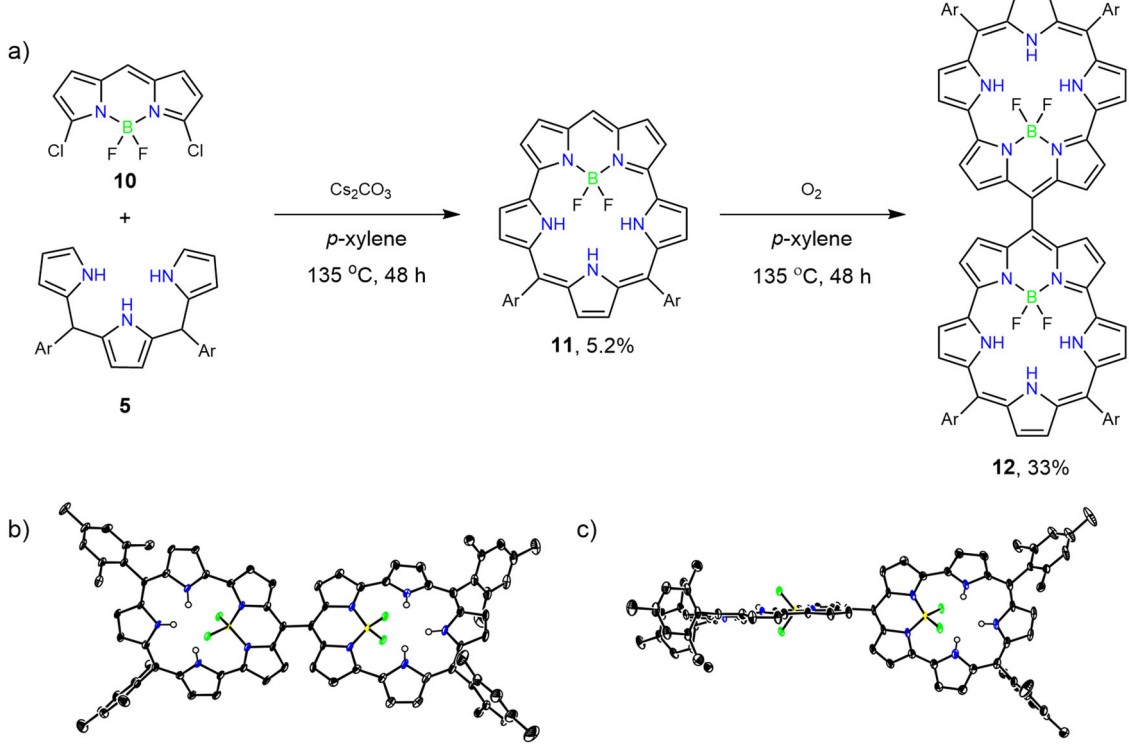

**Fig. 5 Synthesis and X-ray crystal structures of 12. a** Synthesis of **12** through **11**. **b** Top view of crystal structure of **12**, and **c** side view of crystal structure of **12**. Solvent molecules and hydrogen atoms on carbon have been omitted for clarity.

character arising from its 22π-circuit. The meso-free smaragdyrin BF2-complex **11** was isolated in a pure form.

In the meanwhile, however, we found that *meso*-free BF₂-smaragdyrin complex **11** was not stable and slowly oxidized to *meso–meso*-linked BF₂-smaragdyrin dimer **12** in the air. Actually, this oxidative coupling was accelerated at high temperature. After

some experimentations, we found that the best yield of **12** (34%) was reproducibly accomplished by heating a *p*-xylene solution of **11** at 135 °C in the air for 48 h. The ¹H NMR spectrum of **12** in CDCl₃ is simple, displaying five signals due to the outer pyrrolic protons between 10.44 and 9.06 ppm, a singlet due to the inner NH proton at −7.29 ppm, and a triplet due to the inner NH

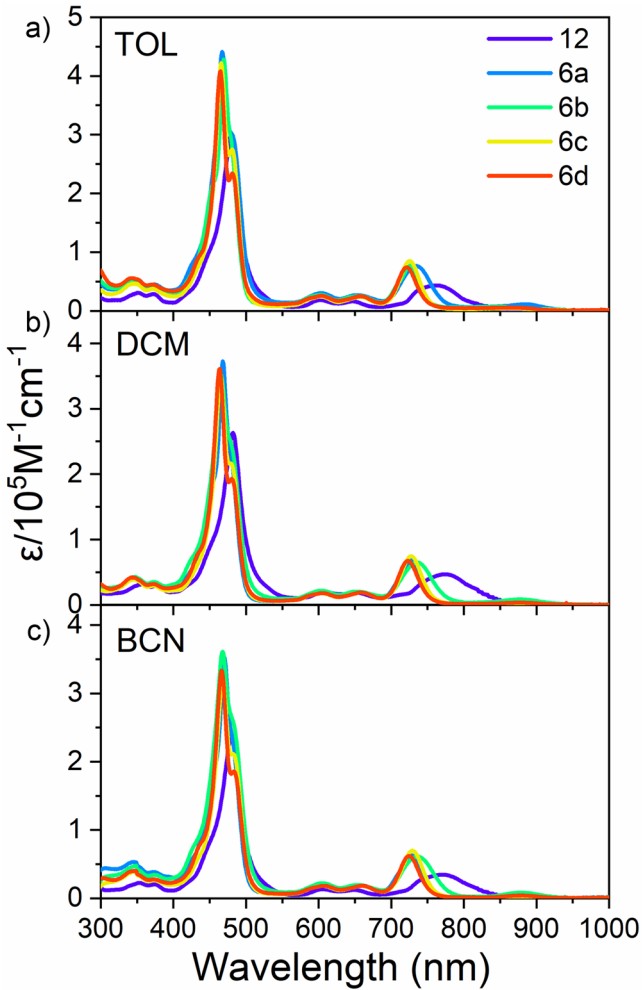

**Fig. 6 Normalized steady-state absorption spectra.** 12 (purple), 6a (blue), 6b (green), 6c (yellow), and 6d (red) in toluene (**a**), in dichloromethane (**b**), in benzonitrile (**c**).

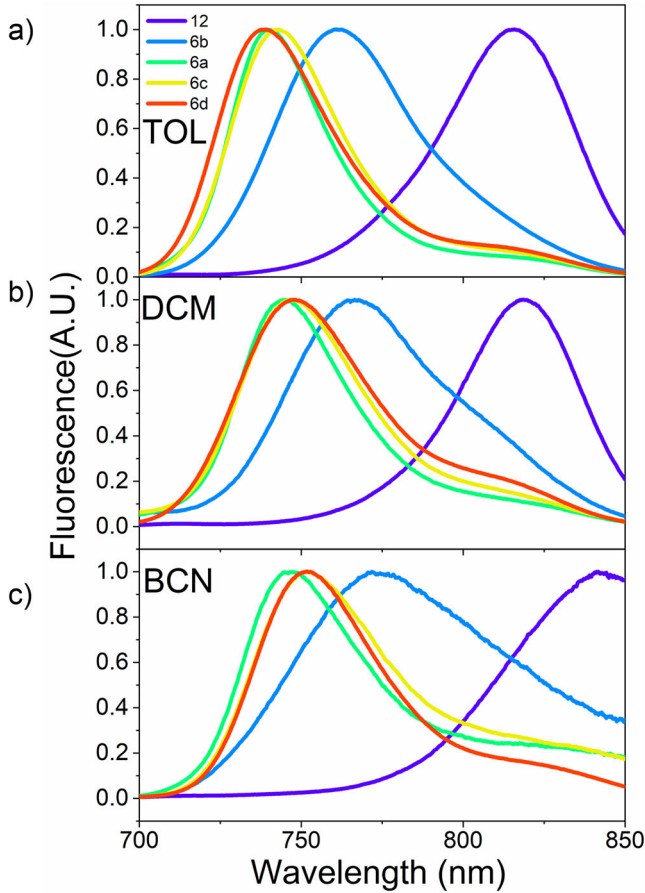

**Fig. 7 Normalized steady-state fluorescence spectra of BF$_2$–smaragdyrins.** 12 (purple), 6a (green), 6b (blue), 6c (yellow), and 6d (red) in toluene (**a**), in dichloromethane (**b**), and in benzonitrile (**c**).

proton at −4.37 ppm. The disappearance of the *meso*-proton signal of 11 was consistent with the formation of the *meso-meso*-linked dimer. The parent ion peak of 12 was observed at *m/z* = 1292.5397 (calcd for $[C_{82}H_{70}B_2F_4N_{10}]^+$; 1292.5925 ($[M]^+$)) in its high-resolution MALDI-TOF mass spectrum. The structure of 12 was unambiguously confirmed by X-ray diffraction analysis (Fig. 5). The C(*meso*)–C(*meso*) bond length is 1.497(4) Å, which is within the range of typical C–C bond. The dihedral angle of the BF$_2$–smaragdyrin rings is 62.5(1)°. These structural features indicate that conjugative interaction between the two BF$_2$–smaragdyrin units is weak.

**UV/Vis/NIR absorption and fluorescence spectra.** The UV/Vis/NIR spectra of these BF$_2$-smaragdyrins in toluene, CH$_2$Cl$_2$, and benzonitrile are shown in Fig. 6 and the Supplementary Fig. 21–26). The absorption spectrum of 11 in toluene shows weak Q-bands at 596, 649, and 702 nm, which are similar to that of 5,10,19-triaryl-substituted BF$_2$-smaragdyrin. As compared with the absorption spectra of monomer 11, those of dimers 6a–6d are all broader both in the Soret and Q-like band regions, probably owing to exciton coupling between the two BF$_2$–amaragdyrin units. Consistent with this, the most red-shifted Q-like bands are red-shifted in the order of 6d < 6c < 6a < 6b < 12 with decreasing distance of the two BF$_2$–amaragdyrin units. Namely, *meso–meso*-linked dimer 12 exhibits the most red-shifted and broadest Q-like

band at 762 nm. The UV/Vis/NIR absorption spectra of these BF$_2$–smaragdyrins are rather insensitive to solvent polarity.

The normalized fluorescence spectra of these BF$_2$–smaragdyrin dimers are shown in Fig. 7 and Supplementary Figs. 21–26. Their maximum emission wavelengths are all in the near-infrared region. Gradual redshifts are observed in the fluorescence spectra in going from 11, 6d, 6c, and 6a, and a substantial shift are found for 6b, and a bigger shift is observed for 12. The fluorescence quantum yields of the dimers are quite characteristic, decreasing with a decrease in the distance between the BF$_2$–smaragdyrin (Table 1), suggesting that the electronic interaction between the two BF$_2$–smaragdyrins causes additional decaying channel. The fluorescence quantum yields in toluene decrease in the order of 11 (0.138) > 6c (0.079) > 6d (0.054) > 6b (0.032) > 6a (0.028) > 12 (<0.001). The observed practically nonfluorescent behavior of 12 in nonpolar toluene is noteworthy. It is also important that an increase in solvent polarity decreases the fluorescence quantum yield. For example, in the case of 6a, 0.032 (toluene) > 0.019 (CHCl$_3$) > 0.018 (THF) > 0.005 (CH$_2$Cl$_2$) > < 0.001 (acetonitrile). Similar trends were observed for 6b–6d. These data have suggested that BF$_2$–smaragdyrin dimers, despite the identical chromophores in the molecule, undergo intramolecular charge transfer (CT), depending on the solvent polarity and the distance between the two BF$_2$–smaragdyrin units. It is probable that the required asymmetric situation is provided by a fluctuation of polar solvents[17–25]. Stokes shifts became increasingly larger from toluene, dichloromethane, and benzonitrile, suggesting that the relaxations of Franck–Condon states to emitting states involve CT. In particular, the photoexcited-state dynamics of 12 is quite

**Table 1 Photophysical properties of BF$_2$–smaragdyrins in different solvents.**

| | Solvent | $\lambda_{Soret}$ (nm) | $\lambda_{Q-max}$ (nm) | $\lambda_{em}$ (nm) | $E$ (eV)[a] | $\Phi_f$[b] | Stokes shift (cm$^{-1}$) | $k_r$ (s$^{-1}$)[d] | $k_{nr}$ (s$^{-1}$)[d] |
|---|---|---|---|---|---|---|---|---|---|
| **6a** | Toluene | 467 | 728 | 748 | 1.68 | 0.032 | 367 | $1.31 \times 10^8$ | $3.97 \times 10^9$ |
| | THF | 466 | 730 | 750 | 1.68 | 0.018 | 365 | – | – |
| | Chloroform | 468 | 731 | 752 | 1.67 | 0.019 | 382 | – | – |
| | DCM | 467 | 730 | 750 | 1.68 | 0.005 | 365 | $1.47 \times 10^8$ | $2.93 \times 10^{10}$ |
| | BCN | 470 | 729 | 747 | | 0.003 | 331 | $1.58 \times 10^8$ | $5.25 \times 10^{10}$ |
| | MeCN | 463 | 727 | 746 | 1.68 | <0.001 | 350 | – | – |
| **6b** | Toluene | 471 | 738 | 752 | 1.67 | 0.028 | 252 | $1.49 \times 10^8$ | $5.17 \times 10^9$ |
| | THF | 470 | 742 | 758 | 1.65 | 0.025 | 284 | – | – |
| | Chloroform | 471 | 742 | 757 | 1.66 | 0.015 | 267 | – | – |
| | DCM | 468 | 739 | 757 | 1.66 | 0.005 | 322 | $1.85 \times 10^8$ | $3.69 \times 10^{10}$ |
| | BCN | 468 | 737 | 772 | | 0.002 | 615 | $1.33 \times 10^8$ | $6.65 \times 10^{10}$ |
| | MeCN | 464 | 737 | 750 | 1.67 | <0.001 | 235 | – | – |
| **6c** | Toluene | 466 | 726 | 740 | 1.69 | 0.079 | 261 | $3.50 \times 10^8$ | $4.08 \times 10^9$ |
| | THF | 465 | 730 | 742 | 1.69 | 0.043 | 222 | – | – |
| | Chloroform | 467 | 730 | 744 | 1.68 | 0.047 | 258 | – | – |
| | DCM | 466 | 729 | 743 | 1.69 | 0.018 | 258 | $4.39 \times 10^8$ | $2.40 \times 10^{10}$ |
| | BCN | 465 | 729 | 753 | | 0.010 | 437 | $2.70 \times 10^8$ | $2.68 \times 10^{10}$ |
| | MeCN | 462 | 726 | 737 | 1.70 | 0.013 | 206 | – | – |
| **6d** | Toluene | 464 | 721 | 738 | 1.70 | 0.054 | 319 | $2.30 \times 10^8$ | $4.03 \times 10^9$ |
| | THF | 463 | 723 | 739 | 1.70 | 0.041 | 299 | – | – |
| | Chloroform | 464 | 723 | 739 | 1.70 | 0.037 | 299 | – | – |
| | DCM | 463 | 723 | 740 | 1.70 | 0.028 | 318 | $1.88 \times 10^8$ | $6.53 \times 10^9$ |
| | BCN | 466 | 725 | 751 | | 0.017 | 478 | $1.73 \times 10^8$ | $1.00 \times 10^{10}$ |
| | MeCN | 459 | 720 | 737 | 1.70 | 0.017 | 320 | – | – |
| **11** | Toluene | 454, 476 | 702 | 715 | 1.75 | 0.138 | 259 | – | – |
| | THF | 452, 475 | 702 | 718 | 1.75 | 0.150 | 317 | – | – |
| | Chloroform | 454, 476 | 704 | 717 | 1.75 | 0.109 | 258 | – | – |
| | DCM | 453, 475 | 702 | 717 | 1.75 | 0.117 | 298 | – | – |
| | MeCN | 448, 471 | 700 | 714 | 1.75 | 0.192 | 280 | – | – |
| **12** | Toluene | 480 | 762 | 816 | 1.57 | <0.001 | 868 | – | – |
| | THF | 478 | 766 | –[c] | – | –[c] | – | – | – |
| | Chloroform | 479 | 770 | –[c] | – | –[c] | – | – | – |
| | DCM | 478 | 768 | 819 | – | <0.001 | 811 | – | – |
| | BCN | 480 | 771 | 842 | | <0.001 | 1094 | – | – |

[a]Excitation energy of S$_1$-state determined by the midpoint of the absorption spectrum and the fluorescence spectrum.
[b]Fluorescence quantum yield.
[c]Fluorescence quantum yield is too low to be accurately determined.
[d]The rates of radiative and non-radiative decay.

curious, since it scarcely emits fluorescence and its Stokes shifts are large, being 868 cm$^{-1}$ in toluene, 811 cm$^{-1}$ in CH$_2$Cl$_2$ and 1094 cm$^{-1}$ in benzonitrile. These large Stokes shifts suggested that emitting state might have large CT properties. It is notable that dimer **6b** showed relatively large Stokes shifts of 615 cm$^{-1}$ in benzonitrile. It was thought that the extent of the CT in the emitting states depended on the electronic interactions between the two smaragdyrins. Thus, the emitting state of dimer **12** might have the largest CT character in the series.

We measured the fluorescence lifetime of BF$_2$–smaragdyrin dimers using the time-correlated single-photon counting (TCSPC) method and the results are summarized in Supplementary Note 1, Supplementary Figs. S57 and S58. We representatively selected three solvents, toluene ($\varepsilon = 2.38$, TOL), CH$_2$Cl$_2$ ($\varepsilon = 8.93$, DCM), and benzonitrile ($\varepsilon = 25.93$, BCN). Compared with the fluorescence lifetime of BF$_2$–smaragdyrin dimers in toluene, those in CH$_2$Cl$_2$ and benzonitrile are clearly shortened, showing the favorable formation of CT state in polar solvents. In the cases of **6a**–**6d**, the fluorescence lifetimes in polar solvents become shorter in **6a** and **6b** compared with that of **6d**, intermediate in **6c**. These results indicate stronger CT characters in spatially closer dimers with stronger electronic interactions. The fluorescence lifetimes in benzonitrile are 0.98 ns for **6d** > 0.37 ns for **6c** > 0.19 ns for **6a** > 0.15 ns for **6b**, while that of **12** is 0.37 ns, being longer than those of **6a**–**6c**, suggesting that the emitting state of **12** is considerably different from those of **6a**–**d**.

Furthermore, to explore the excited-state population dynamics, femtosecond transient absorption (fs-TA) measurements were carried out in the region of 450–780 nm using an excitation pulse at 710 nm (Fig. 8, Supplementary Note 2, Supplementary Figs. 59–61). The TA spectra of the BF$_2$–smaragdyrin dimers display ground-state bleaching (GSB) features in 450–500 nm and above 725 nm regions, which correspond to the steady-state absorption spectra, and broad excited-state absorption (ESA) features in 500–700 nm region.

The TA spectral signatures of the BF$_2$–smaragdyrin dimers are similar to those reported by the previous studies on the monomer[6]. No significant spectral changes in the TA spectra were observed except for the decay time constants. Namely, we did not detect charge-separated states. The TA kinetic profiles are well-fitted with triple exponential decay functions. We attributed the initial decay component ($\tau_1$) to the conformational dynamics. In polar solvents, contribution by the $\tau_1$ component increases in the order of **12** < **6a** < **6b** < **6c** which is in accordance with the increased conformational flexibility of the smaragdyrin dimers. Based on the solvation time constants of CH$_2$Cl$_2$ and benzonitrile, we assigned the structural change as the symmetry-breaking charge transfer process. Especially, in **12**, this process is favorable in benzonitrile, which is consistent with the steady-state fluorescence spectra.

Additionally, we found that the $\tau_2$ component corresponded to the fluorescence lifetimes measured by TCSPC. Thus, we

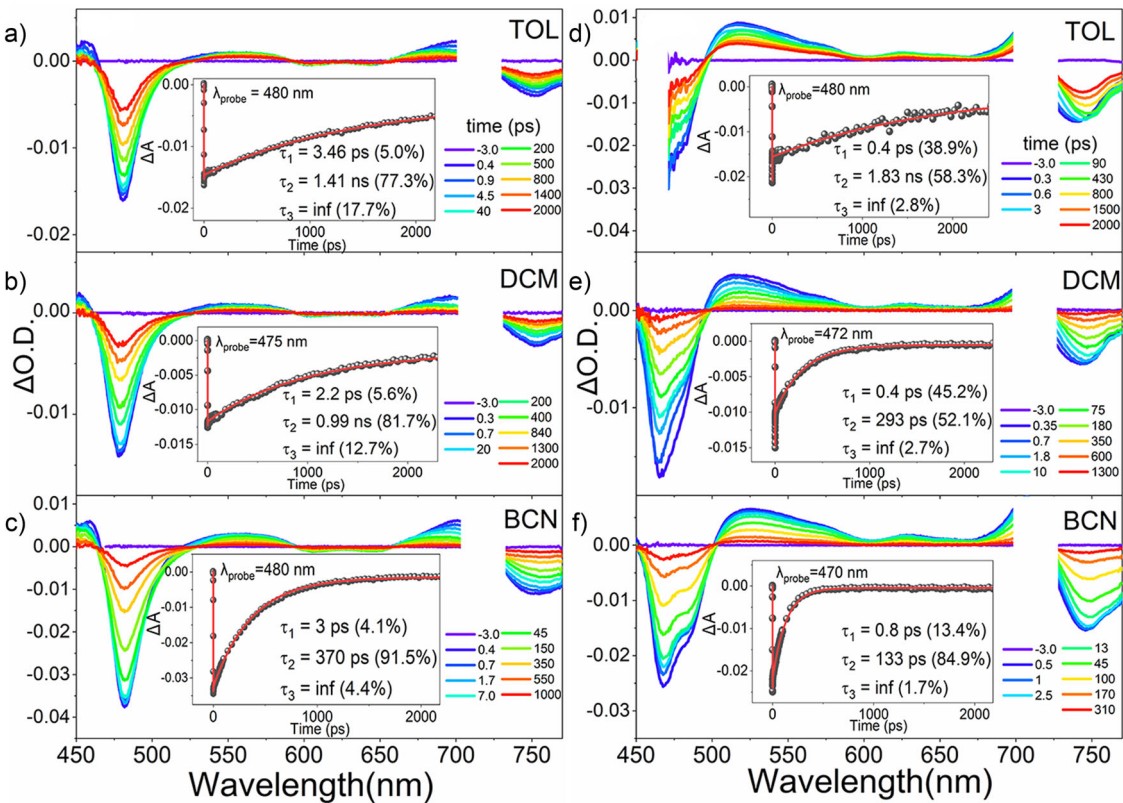

**Fig. 8 Femtosecond-transient absorption (fs-TA) spectra and decay profiles of BF$_2$-Smaragdyrins. 12**, **6b** in toluene (**a**, **d**), in dichloromethane (**b**, **e**), in benzonitrile (**c**, **f**). The samples were pumped at 710 nm with probing visible region to the near-IR region (450–780 nm).

attributed the $\tau_2$ component to the singlet state fluorescence or CT emission decay process, depending on the electronic structures of dimers. These time constants of a series of BF$_2$–smaragdyrin dimers become gradually shortened with an increase of solvent polarity. In the cases of **6a** and **6b**, which have one phenylene bridge, *meta-* and *para-*phenylene linker, respectively, while the spatial separation between the monomer units of **6a** is shorter than **6b**, **6a** exhibits a longer lifetime because the electronic interaction is weaker than **6b**. The major decaying components ($\tau_2$) of **12** were 1.4 ns in toluene, 0.98 ns in CH$_2$Cl$_2$, and 0.37 ns in benzonitrile. These decaying time constants cannot explain the observed very low fluorescent quantum yields of **12**. This means that the emitting state of **12** is not a usual S$_1$-excited state but probably a considerable CT state with a small radiative decaying channel to the ground state. Most probably, the CT state of **12** is formed at a very early time assisted by asymmetric solvation of polar solvent, and decays with a slow time constant to regenerate the ground state. While not a symmetric dimer, Similar accelerated decay of the excited state via charge transfer interaction was reported for a directly linked Zn(II) porphyrin–Zn(II) pheophorbide dyad[28].

Finally, we attributed the $\tau_3$ component to the long-lived triplet state formation through the intersystem crossing. Consequently, the TA spectral features and excited state dynamics unveil that the charge transfer processes, corroborated with the structural evolution according to the structural flexibility and the distance between the monomer units, occur in the BF$_2$–smaragdyrin dimers.

**Electrochemical properties**. Estimation of the free energy gap for the symmetry-breaking charge separation is important to understand the photo-excited dynamics of the smaragdyrin dimers. Thus, we examined their electrochemical properties.

**Table 2 Summary of redox potentials[a].**

| | $E_{ox.1}$/V | $E_{red.1}$/V | $\triangle E_{HL}$/eV[d] | $E_{(S1)}$/V | $\triangle E_{HL} - E_{(S1)}$/eV |
|---|---|---|---|---|---|
| **6a**[b] | 0.02 | −1.66 | 1.68 | 1.67 | 0.01 |
| **6b**[b] | 0.00 | −1.65 | 1.65 | 1.65 | 0.00 |
| **6c**[c] | 0.00 | −1.66 | 1.66 | 1.68 | −0.02 |
| **6d**[b] | −0.02 | −1.66 | 1.64 | 1.70 | −0.04 |
| **12**[b] | −0.06 | −1.62 | 1.56 | 1.56 | 0.00 |
| **6a**[c] | 0.05 | −1.61 | 1.66 | 1.66 | 0 |
| **6b**[c] | 0.05 | −1.60 | 1.65 | 1.65 | 0 |
| **6c**[c] | 0.02 | −1.60 | 1.62 | 1.65 | −0.03 |
| **6d**[c] | 0.02 | −1.59 | 1.61 | 1.61 | 0 |
| **12**[c] | −0.02 | −1.53 | 1.51 | 1.51 | 0 |

[a]Potentials [V] vs. ferrocene/ferrocenium ion. Scan rate, 0.05 Vs$^{-1}$; working electrode, glassy carbon; counter electrode, Pt wire; supporting electrolyte, 0.1 M nBu$_4$NPF$_6$ in CH$_2$Cl$_2$; reference electrode, Ag/AgNO$_3$.
[b]Measured in CH$_2$Cl$_2$.
[c]Measured in benzonitrile.
[d]Electrochemical HOMO-LUMO gaps ($\Delta E_{HL} = e(E_{ox.1} - E_{red.1})$ [eV]).

The electrochemical properties of **6a**–**6d**, and **12** are examined by cyclic voltammetry and differential pulse voltammetry in CH$_2$Cl$_2$ and benzonitrile containing 0.1 M n-Bu$_4$NPF$_6$ as an electrolyte and the results are summarized in Table 2. (For detailed electrochemical data see Supplementary Fig. 27–37, Supplementary Tables 1 and 2) In both solvents, the energy levels of the S1 states and the charge-separated states are almost the same. Considering that the reorganization would be necessary for the charge separation, the symmetry-breaking full charge separation would be difficult. Instead, these data would support the feasibility of partial CT.

**Theoretical calculations**. All calculations were carried out using the Gaussian 09 program. All structures were fully optimized without any symmetry restriction. All geometries were optimized

with the crystal structures as the starting structure at the density functional theory (DFT) method with restricted B3LYP (Becke's three-parameter hybrid exchange functionals and the Lee–Yang–Parr correlation functional) level[19–25]. We calculated the absorption spectra of important compounds, **12**, **6b**, and **6c** (For detailed theoretical calculations data see Supplementary Fig. 54–56, Supplementary Data 9). The calculation results of time-dependent density functional theory (TD-DFT) are in good agreement with the actual absorption spectral data of the compound. The maximum absorption peaks of the Q-like band of these compounds mainly come from electron transition from the highest occupied molecular orbital (HOMO) to the lowest unoccupied molecular orbital (LUMO).

**Summary**. In summary, we have synthesized *meso*-diaryl-substituted BF$_2$–smaragdyrin dimers in low but acceptable yields via the twofold S$_N$Ar reaction and subsequent oxidation reaction with DDQ. We also synthesized *meso*-free and *meso*–*meso* directly linked BF$_2$–smaragdyrin dimer. These molecules were investigated by UV/Vis/NIR absorption, fluorescence measurements, TCSPC method, fs-TA measurements, cyclic voltammetry and theoretical quantum calculations. The excited-state decays of the BF$_2$–smaragdyrin dimers were accelerated with increase of the solvent polarity and a decrease in the distance between the smaragdyrin units. It has been concluded that symmetry-breaking charge transfer plays an important role in the excited-state decays of the smaragdyrin dimers in polar solvents.

## Methods

**Materials and characterization**. $^1$H NMR (500 MHz) and $^{13}$C NMR (126 MHz) spectra were taken on a Bruker ADVANCE-500 spectrometer, and chemical shifts were reported as the delta scale in ppm relative to CHCl$_3$ as an internal reference for $^1$H NMR ($\delta = 7.260$ ppm) and $^{13}$C NMR ($\delta = 77.000$ ppm) (For $^1$H NMR and $^{13}$C NMR spectra see Supplementary Figs. 1–20). UV/Vis absorption spectra were recorded on a Shimadzu UV-3600 spectrometer. Fluorescence emission spectra were recorded on a HITACHI F-4500 spectrometer (For UV/Vis and fluorescence emission spectra see Supplementary Fig. 21–26). MALDI-TOF mass spectra were obtained with a Bruker Ultrafle Xtreme MALDI-TOF/TOF Mass spectrometer and Thermo Scientific Q Exactive ESI Mass spectrometer (For HRMS spectra see Supplementary Figs. 42–53). X-ray crystallographic data were taken on an Agilent SuperNova X-ray diffractometer equipped with a large-area CCD detector. Using Olex2, structures of compounds **6b**, **6c**, **10**, and **12** were solved with the ShelXS structure solution program using Direct Methods and refined with the ShelXL refinement package using least squares minimization. Disordered solvent molecules were treated by the SQUEEZE program of Platon (For single crystal data see Supplementary Figs. 38–41, Supplementary Tables 3–6). Redox potentials were measured by the cyclic voltammetry and differential pulse voltammetry method on an ALS660 electrochemical analyzed model (Solvent: PhCN or CH$_2$Cl$_2$, electrolyte: 0.1 M n-Bu$_4$NPF$_6$, working electrode: glassy carbon, reference electrode: Ag/AgNO$_3$, Counter electrode: Pt wire, scan rate: 0.05 V/s, external reference: ferrocene/ferrocenium cation). Benzonitrile passed through the alumina column was used for electrochemical analysis. Unless otherwise noted, materials obtained from commercial suppliers were used without further purification.

**Synthesis of 4a–4d**. To a solution of corresponding dipyrromethane (1 mmol) in THF (70 mL) at −78 °C, NBS (712 mg, 4 mmol) was added in three portions within 1 h. The mixture was stirred at a low temperature for 3 h. Then DDQ (499 mg, 2.2 mmol) was added to the solution. The mixture was stirred at −78 °C, for 0.5 h and then at room temperature for 3 h. The residue was purified through a long Al$_2$O$_3$ column using THF as an eluent. The solvent was removed in vacuo. Then, CH$_2$Cl$_2$ (100 mL) was added to the remaining solids. TEA (4.0 mL, 28 mmol) and BF$_3$•OEt$_2$ (5.0 mL, 45 mmol) were added dropwise to the solution. The mixture was stirred at room temperature for 24 h. The solvent was removed in vacuo. Products were simply purified by recrystallization with MeOH/H$_2$O. **4a** (280 mg), **4b** (274 mg), **4c** (340 mg) and **4d** (452 mg). **4a**: HR-MS (MALDI-TOF-MS): $m/z = 773.7518$, calcd for (C$_{24}$H$_{12}$B$_2$Br$_4$F$_4$N$_4$)$^+$ = 773.7883 ([M]$^+$). **4b**: HR-MS (MALDI-TOF-MS): $m/z = 773.7731$, calcd for (C$_{24}$H$_{12}$B$_2$Br$_4$F$_4$N$_4$)$^+$ = 773.7883 ([M]$^+$). **4c**: HR-MS (MALDI-TOF-MS): $m/z = 849.7946$, calcd for (C$_{30}$H$_{16}$B$_2$Br$_4$F$_4$N$_4$)$^+$ = 849.8198 ([M]$^+$). **4d**: HR-MS (MALDI-TOF-MS): $m/z = 925.8197$, calcd for (C$_{36}$H$_{20}$B$_2$Br$_4$F$_4$N$_4$)$^+$ = 925.8514 ([M]$^+$).

**Synthesis of 6a–6d**. A solution of brominated BODIPY compound (0.051 mmol), **5** (50 mg, 0.108 mmol), and Cs$_2$CO$_3$ (180 mg, 0.60 mmol) were purged with argon

and then charged with dry *o*-DCB (16 mL). The mixture was stirred at 135 °C for 48 h. The reaction mixture was passed through a short silica gel column using CHCl$_3$ as an eluent and the solvent was evaporated in vacuo. The product was purified by column chromatography on silica gel (CH$_2$Cl$_2$/hexanes, 1:2 V/V) and recrystallization with CH$_2$Cl$_2$/MeOH. **6a** (1.9 mg, 0.0014 mmol, 2.8% yield), **6b** (1.5 mg, 0.0011 mmol, 2.2% yield), **6c** (2.0 mg, 0.0014 mmol, 2.7% yield), and **6d** (1.9 mg, 0.0012 mmol, 2.4% yield) were obtained all as green solids. **6a**: $^1$H NMR (500 MHz, CDCl$_3$): $\delta = 10.49$ (d, 4H, $J = 4.5$ Hz, $\beta$-H), 10.13–10.11 (m, 8H, $\beta$-H), 9.95 (s, 1H, Ar-H), 9.17 (dd, 2H, $J = 7.5$, 1.5 Hz, Ar-H), 9.08 (d, 4H, $J = 1.0$ Hz, $\beta$-H), 8.88–8.87 (m, 4H, $\beta$-H), 8.59 (t, 1H, $J = 7.5$ Hz, $\beta$-H), 7.42 (s, 8H, Ar-H), 2.72 (s, 12H, Me-H), 1.92 (s, 24H, Me-H), −4.44 (t, 4H, $J = 11.5$ Hz, N-H), and −7.37 (s, 2H, N-H) ppm. $^{13}$C NMR (126 MHz, CDCl$_3$) $\delta = 140.1$, 138.4, 138.0, 134.9, 134.4, 132.4, 131.4, 129.8, 128.3, 125.7, 124.6, 123.7, 122.0, 119.6, 118.3, 116.6, 105.8, 21.6, and 21.2 ppm. $\lambda_{max}$ ($\varepsilon$ [M$^{-1}$cm$^{-1}$] in toluene) = 348 (74000), 373 (60000), 467(441000), 604(38000), 658(37000), 728(129000) nm. HR-MS (MALDI-TOF-MS): $m/z = 1368.5771$, calcd for (C$_{88}$H$_{74}$B$_2$F$_4$N$_{10}$)$^+$ = 1368.6239 ([M]$^+$). **6b**: $^1$H NMR (500 MHz, CDCl$_3$): $\delta = 10.59$ (d, 4H, $J = 4.5$ Hz, $\beta$-H), 10.21–10.20 (m, 4H, $\beta$-H), 10.17 (d, 4H, $J = 4.0$ Hz, $\beta$-H), 9.30 (s, 4H, Ar-H), 9.13 (s, 4H, $\beta$-H), 8.95 (br, 4H, $\beta$-H), 7.47 (s, 8H, Ar-H), 2.75 (s, 12H, Me-H), 1.98 (s, 24H, Me-H), −4.34 (t, 4H, $J = 10.5$ Hz, N-H), and −7.27 (s, 2H, N-H) ppm. $^{13}$C NMR (126 MHz, CDCl$_3$) $\delta = 140.1$, 139.2, 138.4, 138.0, 135.0, 134.2, 132.4, 131.4, 129.8, 128.3, 125.7, 124.7, 123.8, 122.0, 119.7, 118.4, 100.0, 21.6, and 21.2 ppm. $\lambda_{max}$ ($\varepsilon$ [M$^{-1}$cm$^{-1}$] in toluene) = 347 (58000), 371 (47000), 471 (428000), 600(32000), 647(28000), 738(105000) nm. HR-MS (MALDI-TOF-MS): $m/z = 1368.5771$, calcd for (C$_{88}$H$_{74}$B$_2$F$_4$N$_{10}$)$^+$ = 1368.6239 ([M]$^+$). **6c**: $^1$H NMR (500 MHz, CDCl$_3$): $\delta = 10.48$ (d, 4H, $J = 4.0$ Hz, $\beta$-H), 10.14–10.13 (m, 4H, $\beta$-H), 9.88 (d, 4H, $J = 4.5$ Hz, $\beta$-H), 9.09 (s, 4H, $\beta$-H), 8.95 (d, 4H, $J = 7.5$ Hz, Ar-H), 8.90 (dd, 4H, $J = 4.0$, 1.5 Hz, $\beta$-H), 8.64 (d, 4H, $J = 7.5$ Hz, Ar-H), 7.45 (s, 8H, Ar-H), 2.74 (s, 12H, Me-H), 1.95 (s, 24H, Me-H), −4.39 (t, 4H, $J = 11.5$ Hz, N-H), and −7.31 (s, 2H, N-H) ppm. $^{13}$C NMR (126 MHz, CDCl$_3$) $\delta = 140.1$, 138.4, 138.0, 135.7, 134.9, 132.2, 131.4, 129.8, 128.3, 127.4, 125.7, 124.6, 123.6, 121.9, 119.5, 118.3, 116.6, 105.8, 21.6, and 21.2 ppm. $\lambda_{max}$ ($\varepsilon$ [M$^{-1}$cm$^{-1}$] in toluene) = 346 (45000), 371 (36000), 466 (422000), 604(24000), 657(23000), 726(80000) nm. HR-MS (MALDI-TOF-MS): $m/z = 1444.6035$, calcd for (C$_{94}$H$_{78}$B$_2$F$_4$N$_{10}$)$^+$ = 1444.6554 ([M]$^+$). **6d**: $^1$H NMR (500 MHz, CDCl$_3$): $\delta = 10.44$ (d, 4H, $J = 4.4$ Hz, $\beta$-H), 10.12–10.11 (m, 4H, $\beta$-H), 9.80 (d, 4H, $J = 4.0$ Hz, $\beta$-H), 9.08 (s, 4H, $\beta$-H), 8.88 (dd, 4H, $J = 4.0$, 1.5 Hz, $\beta$-H), 8.84 (d, 4H, $J = 8.0$ Hz, Ar-H), 8.42 (d, 4H, $J = 7.5$ Hz, Ar-H), 8.32 (s, 4H, Ar-H), 7.44 (s, 8H, Ar-H), 2.73 (s, 12H, Me-H), 1.94 (s, 24H, Me-H), −4.40 (t, 4H, $J = 11.5$ Hz, N-H), and −7.32 (s, 2H, N-H) ppm. $^{13}$C NMR (126 MHz, CDCl$_3$) $\delta = 140.1$, 138.4, 138.0, 135.5, 134.9, 132.1, 131.4, 129.8, 128.3, 128.1, 127.0, 125.7, 124.6, 123.6, 121.8, 119.5, 118.2, 116.61, 116.58, 105.7, 21.6, and 21.2 ppm. $\lambda_{max}$ ($\varepsilon$ [M$^{-1}$cm$^{-1}$] in toluene) = 344 (47000), 371 (34000), 464 (408000), 603(20000), 657(20000), 721(74000) nm. MS (MALDI-TOF-MS): $m/z = 1520.6715$, calcd for (C$_{100}$H$_{82}$B$_2$F$_4$N$_{10}$)$^+$ = 1520.6868 ([M]$^+$).

**Synthesis of 9**. To a slurry of *p*-chloranil (1850 mg, 7.4 mmol) in CH$_2$Cl$_2$ (140 mL) at −40 °C under nitrogen, a solution of **7** (1000 mg, 6.8 mmol) in CH$_2$Cl$_2$ (200 mL) under N$_2$ was added dropwise over several minutes. The reaction mixture was stirred for 3 h. The color of the mixture changed from brown to bright yellow. After DIPEA (7.0 mL, 41 mmol) was added, the solution was stirred for 30 min. BF$_3$•OEt$_2$ (6.8 mL, 61.2 mmol) was then added slowly over several min, and the mixture was stirred for 18 h, during which time the temperature was allowed to rise to 22 °C. The fluorescent solution was sonicated for 30 min and then filtered through a Celite pad to remove insoluble materials. Then, the reaction mixture was washed with saturated aq. NH$_4$Cl and the organic extract was dried over MgSO$_4$. The solvent was removed in vacuo. Purification using silica gel (CH$_2$Cl$_2$/hexanes, 1:1 V/V) gave **9** as dark red solids (488 mg, 38 %). **9**: $^1$H NMR (500 MHz, CDCl$_3$) $\delta = 7.90$ (s, 2H, $\alpha$-H), 7.42 (s, 1H, *meso*-H), 7.15 (d, $J = 4.0$ Hz, 2H, $\beta$-H), and 6.55 (d, $J = 4.0$ Hz, 2H, $\beta$-H) ppm. MS (ESI-MS): $m/z = 193.0739$, calcd for (C$_9$H$_8$BF$_2$N$_2$)$^+$ = 193.0743 ([M + H]$^+$).

**Synthesis of 10**. A mixture of BODIPY **9** (30 mg, 0.156 mmol), CuCl$_2$•2H$_2$O (132.9 mg, 0.78 mmol), and Cu(OTf)$_2$ (282.1 mg, 0.78 mmol) in CH$_3$CN (6 mL) was stirred under air at 80 °C for 20 min. After cooling down to room temperature, the reaction mixture was poured into water, and washed with sat aq. NH$_4$Cl four times, dried over anhydrous Na$_2$SO$_4$, and filtered and evaporated under vacuum. Purification using a silica gel column with an eluent of CH$_2$Cl$_2$/hexanes, 1:1 V/V and recrystallization with n-hexane gave **10** as red solids (17 mg, 55 %). **10**: $^1$H NMR (500 MHz, CDCl$_3$) $\delta = 7.14$ (s, 1H, *meso*-H), 7.09 (d, 2H, $J = 4.0$ Hz, $\beta$-H), and 6.44 (d, 2H, $J = 4.0$ Hz, $\beta$-H) ppm. $^{13}$C NMR (126 MHz, CDCl$_3$) $\delta = 145.8$, 134.1, 131.4, 127.8, and 119.2 ppm. HR-MS (MALDI-TOF-MS): $m/z = 259.9694$, calcd for (C$_9$H$_5$BCl$_2$F$_2$N$_2$)$^+$ = 259.9887 ([M]$^+$).

**Synthesis of 11**. A solution of **10** (30 mg, 0.108 mmol), **5** (54.1 mg, 0.118 mmol), and Cs$_2$CO$_3$ (141.3 mg, 0.432 mmol) was purged with argon, and then charged with *p*-xylene (10 mL). The mixture was stirred at reflux for 48 h. The reaction mixture was passed through a short silica-gel column using CHCl$_3$ as an eluent and the solvent was evaporated in vacuo. The product was purified by column chromatography on silica gel (CH$_2$Cl$_2$/hexanes, 1:4 V/V) and recrystallization with

$CH_2Cl_2$/MeOH, **11** (3.6 mg, 0.0056 mmol, 5.2% yield) was obtained as green solids. Samples can be stored under inert atmosphere environment at low temperature. **11**: $^1$H NMR (500 MHz, CDCl$_3$): $\delta$ = 10.71 (s, 1H, *meso*-H), 10.46 (d, 2H, $J$ = 4.0 Hz, $\beta$-H), 10.16–10.15(m, 2H, $\beta$-H), 9.84 (d, 2H, $J$ = 4.0 Hz, $\beta$-H), 9.14 (s, 2H, $\beta$-H), 8.94 (dd, 2H, $J$ = 4.0, 1.5 Hz, $\beta$-H), 7.44 (s, 4H, Ar-H), 2.73 (s, 6H, Me-H), 1.91 (s, 12H, Me-H), −4.90 (t, 2H, $J$ = 11.0 Hz, N-H), and −7.76 (s, 1H, N-H) ppm. $^{13}$C NMR (126 MHz, CDCl3) $\delta$ = 140.1, 138.5, 138.0, 134.6, 132.6, 131.1, 129.2, 128.3, 125.5, 124.4, 123.2, 121.8, 119.6, 118.5, 105.5, 100.6, 21.6, and 21.2 ppm; HR-MS (MALDI-TOF-MS): $m/z$ = 647.2793, calcd for $(C_{41}H_{36}BF_2N_5)^+$ = 647.3033 $([M]^+)$.

**Synthesis of 12.** A solution of **11** (9 mg, 0.014 mmol) in *p*-xylene (1 mL) was stirred in the air at 135 °C for 48 h. The solvent was evaporated in a vacuo. The product was purified by column chromatography on silica gel ($CH_2Cl_2$/hexanes, 1:2 V/V) and recrystallization with $CH_2Cl_2$/MeOH gave **12** (3.0 mg, 0.0024 mmol, 34% yield) as yellow-green solids. **12**: $^1$H NMR (500 MHz, CDCl$_3$): $\delta$ = 10.41 (d, 4H, $J$ = 4.5 Hz, $\beta$-H), 10.22 (dd, 4H, $J$ = 4.5, 1.5 Hz, $\beta$-H), 9.33 (d, 4H, $J$ = 4.5 Hz, $\beta$-H), 9.23 (s, 4H, $\beta$-H), 9.02 (dd, 4H, $J$ = 4.0, 1.5 Hz, $\beta$-H), 7.49 (s, 8H, Ar-H), 2.76 (s, 12H, Me-H), 2.02 (s, 24H, Me-H), −4.41 (t, 4H, $J$ = 11.5 Hz, N-H), and −7.32 (s, 2H, N-H) ppm. $^{13}$C NMR (126 MHz, CDCl$_3$) $\delta$ = 140.2, 138.5, 138.1, 136.8, 134.9, 131.5, 129.0, 128.3, 125.7, 125.2, 125.1, 122.2, 120.0, 118.9, 112.1, 106.1, 21.6, and 21.3 ppm. $\lambda_{max}$ ($\varepsilon$ [M$^{-1}$cm$^{-1}$] in toluene) =351 (36000), 373 (33000), 480 (301000), 604(20000), 649(19000), 762(52000) nm. HR-MS (MALDI-TOF-MS): $m/z$ = 1292.5397, calcd for $(C_{82}H_{70}B_2F_4N_{10})^+$ = 1292.5925 $([M]^+)$.

## Data availability

The X-ray crystallographic coordinates for structures reported in this study have been deposited at the Cambridge Crystallographic Data Center (CCDC), under deposition numbers 2183007, 2191271, 2183009, 2183010 (**6b**, **6c**, **10**, and **12**). These data can be obtained free of charge from The Cambridge Crystallographic Data Center via www.ccdc.cam.ac.uk/data_request/cif. The data obtained in this study are available within this article and its Supplementary information and are also from the corresponding authors upon reasonable request. Original $^1$H and $^{13}$C spectra, UV/vis absorption spectra and fluorescence spectra, X-ray crystal data, electrochemical data, HR-MS Spectra, TD-DFT calculation results, time-resolved fluorescence decay data, femtosecond transient absorption spectra and decay profiles of the compounds obtained in this manuscript are available in Supplementary information, the cif file of **6b** is available in Supplementary Data 1, and check cif file of **6b** is available in Supplementary Data 2, the cif file of **6c** is available in Supplementary Data 3, and check cif file of **6c** is available in Supplementary Data 4, the cif file of **10** is available in Supplementary Data 5, and check cif file of **10** is available in Supplementary Data 6, the cif file of **12** is available in Supplementary Data 7, and check cif file of **12** is available in Supplementary Data 8, optimized atomic coordinates are available in Supplementary Data 9. The authors declare that all other data supporting the findings of this study are available within the paper and its Supplementary information files.

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

## Acknowledgements

The work is supported by the National Natural Science Foundation of China (Grant nos. 22071052 and 21772036), Science and Technology Innovation Program of Hunan Province (2021RC4059).

## Author contributions

J.S. designed and conducted the project. L.W., X.C., and S.L. performed the synthesis and characterization and measured the optical and electrochemical properties. L.X., Y.R., and S.K. performed X-ray diffraction analysis and DFT calculations. D.K., A.O., and J.S. prepared the paper.

## Competing interests

The authors declare no competing interests.
