## [Peer Review File · Communications Chemistry]

Reviewers' comments:

Reviewer #1 (Remarks to the Author):

Comments on the manuscript number: COMMSCHEM-22-0315-T

The manuscript with the number COMMSCHEM-22-0315-T entitled "Aromatics-Bridged and meso-meso Linked BF₂-Smaragdyrin Dimers: Fast Decays in Polar Solvents by Symmetry-Breaking Charge Transfer" describes the synthesis of four aromatics-bridged and one meso-meso linked BF₂-smaragdyrin dimers. The structural characterizations of these compounds have been studied clearly by MS, NMR, and single crystal X-ray diffraction analysis. Specially, the results of UV-Vis and fluorescence spectra suggested the exist of symmetry-breaking charge transfer in these BF₂-smaragdyrin dimers. I therefore suggest the acceptance of this paper for publication in Communications Chemistry after clarifying the following issues.

- (1) I feel that the language of the text is not native, which worsen the quality of the manuscript.
- (2) There are too many diagrams in the manuscript.
- (3) Please provide 1H-1H COSY NMR spectra and molar extinction coefficients of these new compounds. It would be better if the molar extinction coefficient was used as the ordinate in the UV-Vis absorption spectrum.
- (4) There is a little ambiguity in the following two sentences, please modify them. "The isolation of the meso-free smaragdyrin BF₂-complex 11 was somewhat unexpected, since most of meso-free expanded porphyrins have been known to be chemically unstable owing to the facile oxidation at the free-meso position" and "however, we found that meso-free BF₂-smaragdyrin complex 11 was not stable"
- (5) The ELECTROCHEMICAL PROPERTIES and THERORETICAL CALCULATIONS section can be discussed in more detail around the charge transfer.
- (6) what does E(S1) mean in Table 2?
- (7) The title of CONCLUSION section is "Discussion"?

Reviewer #2 (Remarks to the Author):

Smaragdyrins are an intriguing class of porphyrinoid macrocycles that have been little studied until recently due to the difficulty in their synthesis. In this contribution, Song and coworkers report the synthesis, X-ray structures, electrochemical and photophysical behaviour, and (TD)DFT calculations of new BF₂-smaragdyrin dimers. This combined set of experimental and theoretical studies provide new insights into the optical properties of smaragdyrins. In particular, the focus of this paper is on demonstrating that there is symmetry-breaking charge transfer (SBCT) upon photoexcitation in these systems, a rare observation for porphyrinoid dimers. SBCT is an important process that the community needs to understand better due to its role in e.g. photosynthesis. Therefore, the topic of this paper is timely and is of wider significance.

I have read the synthetic section and the accompanying characterisation data and I am satisfied that the procedures have been reported in a robust and reproducible way and that the data is consistent.

Furthermore, I have read with interest the detailed photophysical study, and again think that the authors have conducted all the necessary spectroscopic measurements to support their arguments, including detailed TCSPC and fs transient absorption studies with solvent dependence.

It is for these reasons, and in particular the careful approach the authors have taken in their photophysical studies, that I recommend publication of this article essentially as is.

My only minor suggestions are:

- (1) the first sentence of the abstract isn't necessary as it doesn't concern the current paper and this historical note is covered again in the opening paragraph of the introduction.
- (2) yields could be included in figures 2, 4 and 6.
- (3) correct Fig 4 caption to: 'Solvent molecules and hydrogen atoms on carbon...'
- (4) the lifetime fits for the TCSPC studies should be added to the summary of photophysical data in Table 1, which together with the quantum yields of fluorescence can be used to calculate the rates of radiative and non-radiative decay to allow easier comparison between the data in different solvents.
- (5) The section called discussion is a little curious as it is just a summary - were the authors intending this as a conclusion which otherwise seems to be missing?

Reviewer #3 (Remarks to the Author):

The manuscript presents the synthesis of aromatics-bridged smaragdyrin dimers, meso-free BF₂-smaragdyrin, and its meso-meso linked BF₂-smaragdyrin dimer. The macrocycles have been extensively studied with the help of NMR, absorption spectroscopy and fluorescence spectroscopy. Song, Osuka and coworkers have pioneered the work on aza-smaragdyrin macrocycles and published several interesting papers by using the same synthetic strategy. However, in my opinion this manuscript lacks the novelty and quality to be published in a high impact journal like the Communications Chemistry. The photophysical properties are interesting but doesn't warrant publication in Communications Chemistry. This report seems like a mere extension of their previous reports on all aza-smaragdyrins. Hence I don't recommend the publication of the manuscript in Communications Chemistry

Response to Reviewer 1

Reviewer 1 wrote: The manuscript with the number COMMSCHEM-22-0315-T entitled "Aromatics-Bridged and meso-meso Linked BF₂-Smaragdyrin Dimers: Fast Decays in Polar Solvents by Symmetry-Breaking Charge Transfer" describes the synthesis of four aromatics-bridged and one meso-meso linked BF₂-smaragdyrin dimers. The structural characterizations of these compounds have been studied clearly by MS, NMR, and single crystal X-ray diffraction analysis. Specially, the results of UV-Vis and fluorescence spectra suggested the exist of symmetry-breaking charge transfer in these BF₂-smaragdyrin dimers. I therefore suggest the acceptance of this paper for publication in Communications Chemistry after clarifying the following issues.

Response: We thank the high evaluation of **Reviewer 1**.

(1) **Reviewer 1 wrote:** I feel that the language of the text is not native, which worsen the quality of the manuscript.

Response: We polished the language again

(2) There are too many diagrams in the manuscript.

Response: Please understand many diagrams are necessary for this paper.

(3) **Reviewer 1 wrote:** Please provide ¹H-¹H COSY NMR spectra and molar extinction coefficients of these new compounds. It would be better if the molar extinction coefficient was used as the ordinate in the UV-Vis absorption spectrum.

Response: We added ¹H-¹H COSY NMR spectra and molar extinction coefficients of all new compounds. Molar extinction coefficient is used as the ordinate in the UV-Vis absorption spectrum in the revised manuscript.

(4) There is a little ambiguity in the following two sentences, please modify them. "The isolation of the meso-free smaragdyrin BF₂-complex 11 was somewhat unexpected, since most of meso-free expanded porphyrins have been known to be chemically unstable owing to the facile oxidation at the free-meso position" and "however, we found that meso-free BF₂-smaragdyrin complex 11 was not stable"

Response: We rewrote these two sentences to "The meso-free smaragdyrin BF₂-

complex **11** was isolated in a pure form. In the meanwhile, we found that meso-free BF₂-smaragdyrin complex **11** was not stable”.

(5) The ELECTROCHEMICAL PROPERTIES and THERORETICAL CALCULATIONS section can be discussed in more detail around the charge transfer.

Response: We rewrote as following:

In both solvents, the energy levels of the S1-states and the charge-separated states are almost the same. Considering that the reorganization would be necessary for the charge separation, the symmetry-breaking full charge separation would be difficult. Instead, these data would support the feasibility of partial CT.

(6) what does E(S1) mean in Table 2?

Response: E(S1) in Table 2 means the excitation energy of S1-state of smaragdyrins.

(7) The title of CONCLUSION section is “Discussion”?

Response: The “Discussion” section should be “Summary”, we revised this in the revised version.

Response to Reviewer 2

Smaragdyrins are an intriguing class of porphyrinoid macrocycles that have been little studied until recently due to the difficulty in their synthesis. In this contribution, Song and coworkers report the synthesis, X-ray structures, electrochemical and photophysical behaviour, and (TD)DFT calculations of new BF₂-smaragdyrin dimers. This combined set of experimental and theoretical studies provide new insights into the optical properties of smaragdyrins. In particular, the focus of this paper is on demonstrating that there is symmetry-breaking charge transfer (SBCT) upon photoexcitation in these systems, a rare observation for porphyrinoid dimers. SBCT is an important process that the community needs to understand better due to its role in e.g. photosynthesis. Therefore, the topic of this paper is timely and is of wider significance.

I have read the synthetic section and the accompanying characterisation data and I am

satisfied that the procedures have been reported in a robust and reproducible way and that the data is consistent. Furthermore, I have read with interest the detailed photophysical study, and again think that the authors have conducted all the necessary spectroscopic measurements to support their arguments, including detailed TCSPC and fs transient absorption studies with solvent dependence.

It is for these reasons, and in particular the careful approach the authors have taken in their photophysical studies, that I recommend publication of this article essentially as is.

Response: We thank the high evaluation of **Reviewer 2**.

(1) **Reviewer 2 wrote:** the first sentence of the abstract isn't necessary as it doesn't concern the current paper and this historical note is covered again in the opening paragraph of the introduction.

Response: The first sentence is removed.

(2) **Reviewer 2 wrote:** yields could be included in figures 2, 4 and 6.

Response: Yields were added in figures 2, 4 and 6.

(3) **Reviewer 2 wrote:** correct Fig 4 caption to: 'Solvent molecules and hydrogen atoms on carbon...'

Response: Fig 5 caption has been corrected. (Reviewer 2 probably means Fig 5 caption)

(4) **Reviewer 2 wrote:** the lifetime fits for the TCSPC studies should be added to the summary of photophysical data in Table 1, which together with the quantum yields of fluorescence can be used to calculate the rates of radiative and non-radiative decay to allow easier comparison between the data in different solvents.

Response: We added the rates of radiative and non-radiative decay to table 1

(5) **Reviewer 2 wrote:** The section called discussion is a little curious as it is just a summary - were the authors intending this as a conclusion which otherwise seems to be missing?

Response: The "Discussion" section should be "Summary", we revised this in the revised version.

REVIEWERS' COMMENTS:

Reviewer #1 (Remarks to the Author):

Since the authors have clarified almost all the issues concerned, I suggest the acceptance of this paper for publication in this journal in the present form.

Reviewer #2 (Remarks to the Author):

The authors have satisfactorily addressed all of the reviewers' comments, and thus the manuscript can be accepted as is in this revised form.